# Acute paracoccidioidomycosis worsened by immunosuppressive therapy due to a misdiagnosis of Crohn's disease

**Eduardo Mastrangelo Marinho Falcão**[1]*, **Manuela da Costa Medeiros**[2], **Andrea d´Ávila Freitas**[2], **João Carlos de Almeida Soares**[3], **Maria Inês Fernandes Pimentel**[4], **Leonardo Pereira Quintella**[5], **Dayvison Francis Saraiva Freitas**[1], **Priscila Marques de Macedo**[1], **Antônio Carlos Francesconi do Valle**[1]

**1** Clinical Research Laboratory on Infectious Dermatology, Evandro Chagas National Institute of Infectious Diseases, Fiocruz, Rio de Janeiro, Brazil, **2** Department of Inpatient Health Care, Evandro Chagas National Institute of Infectious Diseases, Fiocruz, Rio de Janeiro, Brazil, **3** Department of Gastroenterology, Evandro Chagas National Institute of Infectious Diseases, Fiocruz, Rio de Janeiro, Brazil, **4** Laboratory of Clinical Research and Surveillance on Leishmaniasis, Evandro Chagas National Institute of Infectious Diseases, Fiocruz, Rio de Janeiro, Brazil, **5** Anatomical Pathology Service, Evandro Chagas National Institute of Infectious Diseases, Fiocruz, Rio de Janeiro, Brazil

\* eduardo.falcao@ini.fiocruz.br

**Data Availability Statement:** All relevant data are within the manuscript.

## Abstract

Paracoccidioidomycosis (PCM) is a systemic mycosis endemic in Latin America, mostly in Brazil. The involvement of the gastrointestinal tract is uncommon and usually associated with the acute form. Recently, a cluster of acute PCM cases has been described in Rio de Janeiro, Brazil. We report a 42-year-old male, resident of Rio de Janeiro, presenting chronic diarrhea and abdominal pain in the past 3 years, previously diagnosed as Chron´s disease. When immunosuppressive therapy was prescribed, the patient evolved with worsening of the previous symptoms in addition to odynophagia, 20 kg-weight loss, disseminated skin lesions, diffuse lymphadenopathy and adrenal insufficiency. Histopathological and mycological examination of a skin lesion were compatible with PCM. Itraconazole was prescribed in high doses (400 mg/day). After seven months of treatment, the patient presented with acute abdominal pain which led to an emergent appendectomy, revealing the presence of the fungus. After 24 months, the patient reached clinical cure and recovered from adrenal insufficiency. We emphasize the importance of PCM as a differential diagnosis in patients with chronic diarrhea. The risk of fungal infections should be considered prior to initiating immunosupressive therapies, particularly in endemic areas.

## Author summary

Paracoccidioidomycosis is a systemic mycosis caused by inhalation of fungi belonging to the genus *Paracoccidioides*. It is endemic in Latin America, mainly affects rural workers and is still a neglected disease. In most cases, it affects multiple organs at the same time, mainly lungs, skin and mucosa of the upper aerodigestive tract, while involvement of the

**Funding:** The authors received no specific funding for this work.

**Competing interests:** The authors have no competing interests.

gastrointestinal tract is rare. We report a case misdiagnosed as inflammatory bowel disease (Crohn´s disease). The immunosuppressive therapy led to the worsening of existing symptoms, to the involvement of other organs and then the correct diagnosis of PCM was revealed. The reported case demonstrates the importance of early diagnosis of this mycosis and the risk of starting immunosuppressive therapy in these patients. Furthermore, the place of residence and period in which the patient presented the disease coincides with recent changes in the epidemiology of the disease in the state of Rio de Janeiro, Brazil.

## Presentation of case

A 42-year-old previously healthy male resident in the countryside of Rio de Janeiro state was admitted to our institute presenting with ulcerated lesions on the eyebrows, nasal skin and mucosa, and generalized lymphadenopathy (Fig 1). He had chronic diarrhea and abdominal pain for the past 3 years, which had been recently diagnosed as Crohn´s disease (CD) through colonoscopy (Fig 2) and computed tomography enterography that revealed erosions and a vegetative lesion in the cecum. Treatment with mesalazine (2,400 mg/day) and prednisone (1.0 mg/kg/day) was prescribed, but his clinical condition progressively worsened in a few months, evolving with odynophagia, a 20 kg weight loss, disseminated skin lesions and diffuse lymphadenopathy.

The histopathological examination of the nasal skin lesion showed a chronic diffuse suppurative granulomatous dermatitis with numerous fungal forms compatible with *Paracoccidioides* spp. better seen in the silver staining (Fig 3). Direct examination showed large yeast cells with multiple budding, culture at 25˚C yielded a filamentous colony compatible with *Paracoccidioides* spp. which was successfully converted to the yeast phase at 37oC, and serum anti-*Paracoccidioides* antibodies were detected by double immunodiffusion test (DID) (titer 1:256). Serologies for histoplasmosis, aspergillosis and HIV were negative. Abdominal ultrasound revealed multiple lymphadenopathies and ACTH stimulation test showed adrenal insufficiency. Since the patient was clinically stable and refused hospitalization, we decided to start a high dose of oral itraconazole (400 mg/day) to ensure optimal absorption of this antifungal. Prednisone 7.5 mg/day was prescribed for suspected paracoccidioidomycosis-related

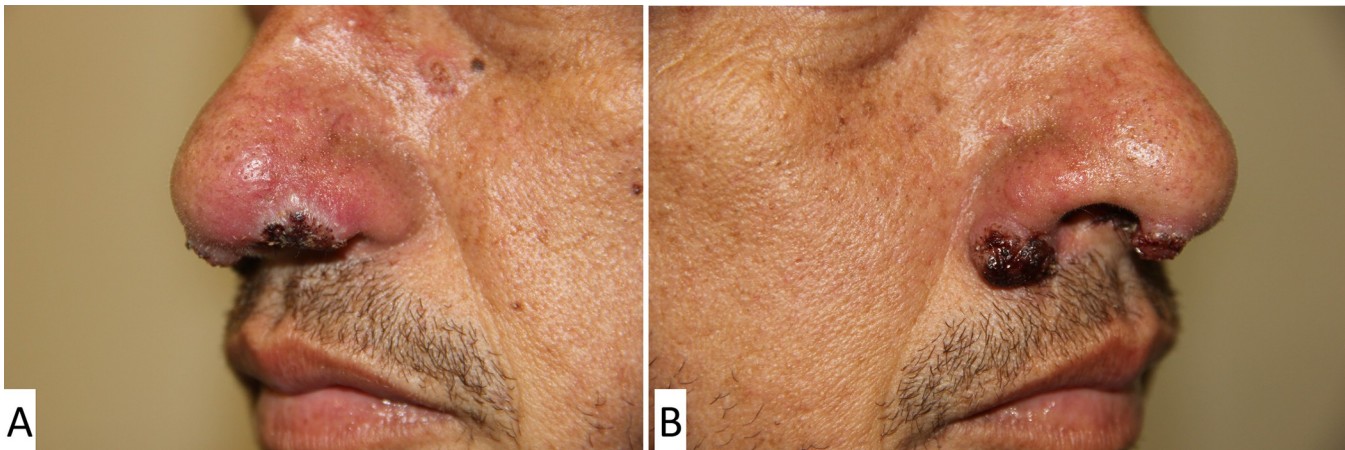

**Fig 1. Ulcerated lesions on the nasal skin and mucosa.**

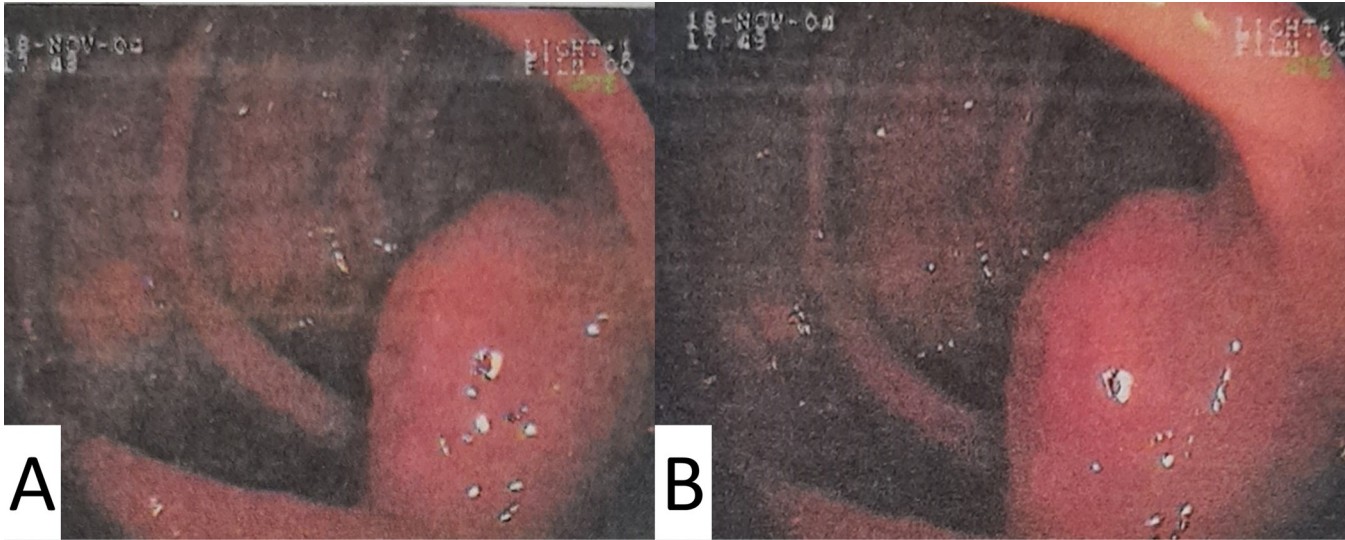

**Fig 2. Colonoscopy: Vegetating lesion in the cecum with an irregular and friable surface.**

adrenal insufficiency. The skin lesions and lymphadenopathies gradually reduced in number and size despite persistent intestinal disease. After seven months of treatment, the patient presented with acute abdominal pain, which led to an emergent appendectomy. The histopathological examination of the appendix (Fig 4) identified the presence of *Paracoccidioides* spp. structures. Treatment continued for up to 24 months, when the patient reached clinical cure including recovery from gastrointestinal symptoms, skin and mucosal lesions and adrenal insufficiency, negative serological DID titers, and no evidence of disease on CT enterography and a repeat Colonoscopy revealed healed lesions without evidence of fungal organisms on histopathological examination. He was followed for two years after the end of treatment, remaining cured without complications or sequelae.

## Case discussion

Paracoccidioidomycosis (PCM) is a systemic mycosis endemic in Latin America, mostly in Brazil, where around 80% of the cases occur [1]. Thermodimorphic fungi of the genus

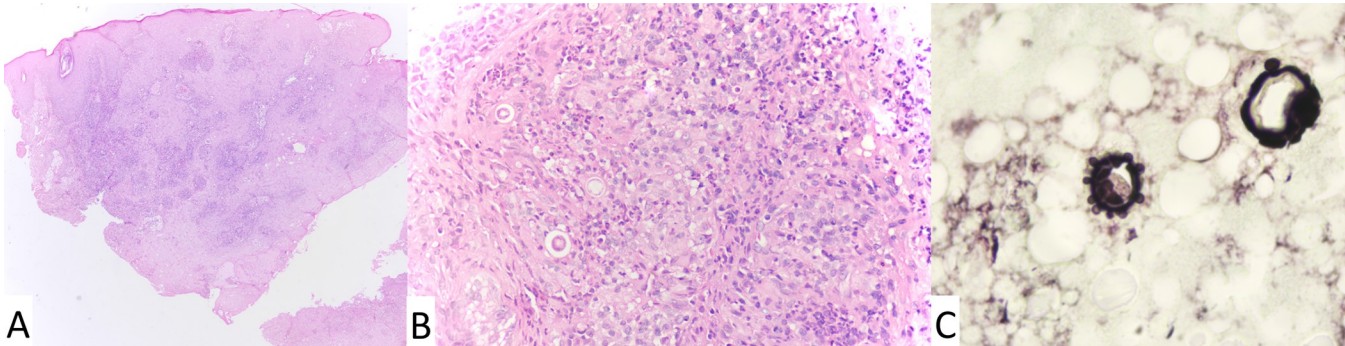

**Fig 3. Histopathological examination of the nasal skin.** (A) Diffuse dermatitis with pseudoepitheliomatous hyperplasia (HE, 40x); (B) Suppurative granuloma with refringent round fungal forms (HE, 400x); (C) Round structures with multiple budding ("steering wheels") (Grocott 1000x).

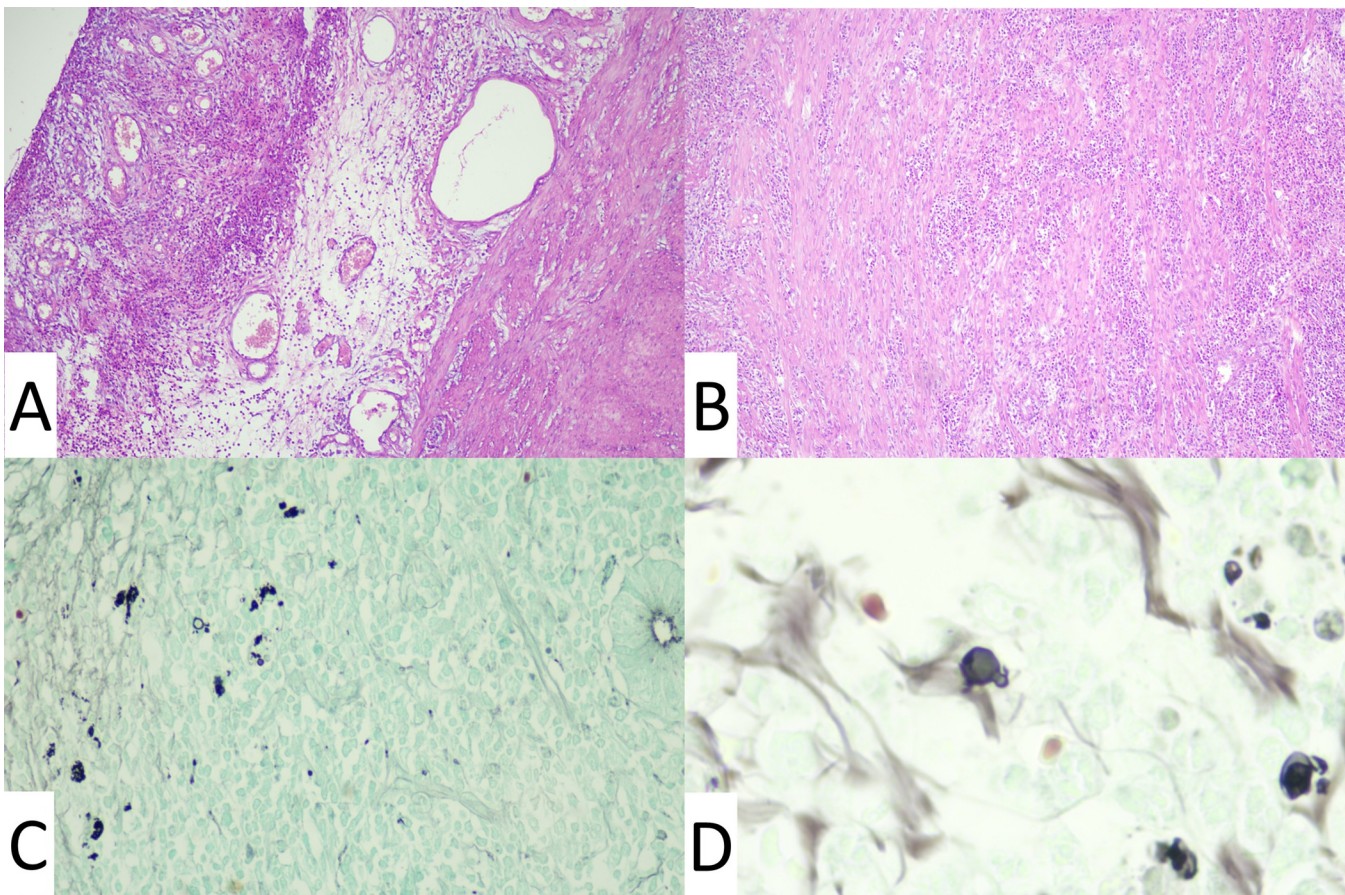

**Fig 4. Histopathological examination of the cecal appendix. Acute appendicitis associated with paracoccidioidomycosis.** (A) Mixed inflammatory infiltrate, rich in neutrophils, permeating the muscle wall (HE, 100x); (B) Hyperemia, edema, mixed inflammatory infiltrate and fibrinoid exudate in the serous layer (HE, 100x); (C) Round fungal form with single budding and numerous collapsed or fragmented fungal forms (left) and Lieberkühn's crypt (right) (Grocott, 400x); (D) Round fungal form with single budding and fragmented or deformed fungal forms (Grocott, 1000X).

*Paracoccidioides* are the etiological agents and infection occurs through inhalation of their infectious propagules after activities related to the management of the soil. The disease can manifest acutely or after a long latency period, characterizing respectively, the acute (juvenile type) and the chronic (adult type) forms [2].

The acute form is less frequent but usually more severe due to an immunological susceptibility of the host induced by a genetic predisposition or other immunosuppressive conditions, mostly HIV/AIDS [3]. In this clinical form, the reticuloendothelial system and intraabdominal organs are usually affected. The involvement of the gastrointestinal tract is uncommon and can be the only affected organ in rare PCM cases, which may represent a diagnostic challenge [4]. The gastrointestinal involvement in PCM is probably underdiagnosed as Peyer's patches, as part of the reticuloendothelial system, are expected to be frequently achieved and endoscopic examination is not routinely performed in PCM patients.

PCM mimicking CD has been scarcely reported in the literature and in most reported cases the diagnosis came after several months of symptoms [5–8]. The wrong diagnosis led to immunosuppressive therapy leading to worsening of symptoms in some cases [6–8]. A recently published systematic review found 46 cases of gastrointestinal PCM predominantly in young men presenting the acute PCM form. The most common signs and symptoms observed were

diarrhea, abdominal pain and weight loss. The large intestine was most frequently affected (78.2%), followed by the small intestine (52.1%). Noteworthy, a CD clinical behavior was described in 10 among the 46 cases (21.7%) while the presence of both PCM and inflammatory bowel disease (IBD) occurred in only one case [5]. In the case herein reported, although we observed the most frequently reported gastrointestinal clinical signs, other findings such as cutaneous lesions and generalized lymphadenopathy occurred after initiation of immunosuppressive drugs to treat CD. Even though the cutaneous involvement occurred in the context of clinical worsening, it allowed the prompt diagnosis of PCM as these lesions were of easy access.

The colonoscopy findings of PCM may include ulcers clinically similar to CD [6–8] and the histopathological study of these lesions may be unspecific or difficult to differentiate if yeast-like fungal structures of *Paracoccidioides* spp. are not visualized since granulomatous inflammation can be observed in both diseases [5,9]. In the present case, clinical and endoscopic features led to a misdiagnosis of CD.

Direct mycological examination is a low-cost test that may provide a rapid diagnosis of PCM, but some limitations include its limited sensitivity and the availability of clinical specimens, not always easy to access. In this context, serological tests play an important role in cases without diagnostic confirmation, although they are not universally accessible [2].

Treatment of PCM with intravenous antifungals (primarily amphotericin B in deoxycholate at doses of 0.5–0.7mg/kg/day or in lipid formulation at doses of 3.0–5.0 mg/kg/day) is usually indicated in gastrointestinal involvement because malabsorption can occur in these cases [2]. In addition, surgery is frequently necessary for PCM complications such as intestinal obstruction related to expansive or scarring lesions [5]. The patient herein reported refused hospitalization but presented a good response to higher doses of oral itraconazole. Although he had to undergo an appendectomy after seven months of treatment, he did not have any further complications.

Considering that epidemiological changes of PCM have been taking place over the years, health professionals should be aware that this fungal disease, especially in its acute form, must be included in the differential diagnosis of IBD. In Rio de Janeiro state, Brazil, a cluster of acute PCM cases was described during a highway construction in a period in which there was also clearing of forests, soil humidity, and the El Niño phenomenon. The incidence increased from 1.29 to 8.25 cases per million persons in the affected area exposing in most cases the low-income population that lives in Baixada Fluminense. [9]. The patient reported was probably affected as he lives 2.3 km from the aforementioned highway.

The occurrence of PCM in urban areas exposes immunocompromised patients to a higher risk of infection and they may present disease with higher severity [3]. The misdiagnosis of PCM as CD represents an additional risk because the immunosuppressive therapy for CD modifies the natural history of PCM, reactivating latent foci and/or aggravating an active disease as drug-induced immunosuppression leads to a greater bloodstream spread of the fungal agent, increasing the risk of complications, sequelae, and deaths [10].

As PCM is usually a multifocal disease [1,2], multidisciplinary care is required for the proper diagnosis and clinical management, including dermatologists, gastroenterologists, surgeons, pathologists, mycologists, otorhinolaryngologists and neurologists.

Other infectious diseases including endemic systemic mycoses such as histoplasmosis [11] may present initially with gastrointestinal symptoms and should be considered in the differential diagnosis of these patients before initiating immunosuppressive therapy. Early clinical suspicion, laboratory diagnosis and specific treatment can help to prevent poor outcomes.

## Key learning points

- PCM and other fungal infections should be considered in the differential diagnosis in patients with chronic diarrhea, particularly in endemic areas.

- The risk of fungal infections (or reactivations) should be considered prior to initiating immunosuppressive therapies.

- Patients with PCM require a multidisciplinary clinical management.

## Acknowledgments

The Evandro Chagas National Institute of Infectious Diseases (INI/FIOCRUZ) which provided infrastructure needed for diagnosis and treatment of the patient.

## Ethics statement

The Research Ethics Committee of the Evandro Chagas National Institute of Infectious Diseases (INI/FIOCRUZ), a reference center for PCM in Rio de Janeiro State, Brazil approved this study (appreciation number 26066619.0.0000.5262), and a written formal consent was obtained from the patient.

## Author Contributions

**Conceptualization:** Eduardo Mastrangelo Marinho Falcão, Manuela da Costa Medeiros, Maria Inês Fernandes Pimentel, Dayvison Francis Saraiva Freitas, Priscila Marques de Macedo, Antônio Carlos Francesconi do Valle.

**Data curation:** Eduardo Mastrangelo Marinho Falcão, Manuela da Costa Medeiros, Andrea d´Ávila Freitas, Maria Inês Fernandes Pimentel, Leonardo Pereira Quintella, Dayvison Francis Saraiva Freitas, Priscila Marques de Macedo, Antônio Carlos Francesconi do Valle.

**Formal analysis:** Eduardo Mastrangelo Marinho Falcão, Manuela da Costa Medeiros, Dayvison Francis Saraiva Freitas, Priscila Marques de Macedo, Antônio Carlos Francesconi do Valle.

**Investigation:** Eduardo Mastrangelo Marinho Falcão, Manuela da Costa Medeiros, Maria Inês Fernandes Pimentel, Leonardo Pereira Quintella, Dayvison Francis Saraiva Freitas, Priscila Marques de Macedo, Antônio Carlos Francesconi do Valle.

**Methodology:** Andrea d´Ávila Freitas, João Carlos de Almeida Soares, Maria Inês Fernandes Pimentel, Leonardo Pereira Quintella, Dayvison Francis Saraiva Freitas, Priscila Marques de Macedo, Antônio Carlos Francesconi do Valle.

**Project administration:** Priscila Marques de Macedo, Antônio Carlos Francesconi do Valle.

**Resources:** João Carlos de Almeida Soares, Leonardo Pereira Quintella, Antônio Carlos Francesconi do Valle.

**Supervision:** Andrea d´Ávila Freitas, Leonardo Pereira Quintella, Priscila Marques de Macedo, Antônio Carlos Francesconi do Valle.

**Writing – original draft:** Eduardo Mastrangelo Marinho Falcão, Manuela da Costa Medeiros, Andrea d´Ávila Freitas, João Carlos de Almeida Soares, Maria Inês Fernandes Pimentel, Leonardo Pereira Quintella, Dayvison Francis Saraiva Freitas, Priscila Marques de Macedo, Antônio Carlos Francesconi do Valle.

**Writing – review & editing:** Eduardo Mastrangelo Marinho Falcão, Manuela da Costa Medeiros, Andrea d´Ávila Freitas, João Carlos de Almeida Soares, Maria Inês Fernandes Pimentel, Leonardo Pereira Quintella, Dayvison Francis Saraiva Freitas, Priscila Marques de Macedo, Antônio Carlos Francesconi do Valle.

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
