## [Decision Letter · Decision Letter 0]

23 Nov 2022

Dear Dr Marinho Falcão,

Thank you very much for submitting your manuscript "Acute paracoccidioidomycosis worsened by immunosuppressive therapy due to a misdiagnosis of Crohn’s disease" for consideration at PLOS Neglected Tropical Diseases. As with all papers reviewed by the journal, your manuscript was reviewed by members of the editorial board and by several independent reviewers. The reviewers appreciated the attention to an important topic. Based on the reviews, we are likely to accept this manuscript for publication, providing that you modify the manuscript according to the review recommendations. 

Sincerely,

Joshua Nosanchuk, MD

Section Editor

Joshua Nosanchuk

Section Editor

Reviewer's Responses to Questions

**Key Review Criteria Required for Acceptance?**

**Methods**

-Are the objectives of the study clearly articulated with a clear testable hypothesis stated?

-Is the study design appropriate to address the stated objectives?

-Is the population clearly described and appropriate for the hypothesis being tested?

-Is the sample size sufficient to ensure adequate power to address the hypothesis being tested?

-Were correct statistical analysis used to support conclusions?

-Are there concerns about ethical or regulatory requirements being met?

Reviewer #1: methods n/a in this type of article.

Reviewer #2: (No Response)

**Results**

-Does the analysis presented match the analysis plan?

-Are the results clearly and completely presented?

-Are the figures (Tables, Images) of sufficient quality for clarity?

Reviewer #1: Figures are adequate.

Reviewer #2: (No Response)

**Conclusions**

-Are the conclusions supported by the data presented?

-Are the limitations of analysis clearly described?

-Do the authors discuss how these data can be helpful to advance our understanding of the topic under study?

-Is public health relevance addressed?

Reviewer #1: The first key learning point is supported in the article. 

The second key learning point should be reworded. The phrase "before excluding fungal infections" almost implies that fungal infections are routinely screened prior to initiation of immunosuppressive agents. Consider something along the line of... "The risk of fungal infections (or reactivations) should be considered prior to initiating immunosuppressive therapies." 

The third key learning point did not come through in the article. It was not clear to me where the multidisciplinary aspect was discussed.

Reviewer #2: (No Response)

**Editorial and Data Presentation Modifications?**

Reviewer #1: Needs revisions, though largely minor, would not accept as is.

Reviewer #2: (No Response)

**Summary and General Comments**

Reviewer #1: This article does add to the cases of endemic fungi being misdiagnosed as Crohn's disease and therefore should be an accepted paper. I enjoyed the case and appreciated the figures.

What was clear to me is that there were (at least) two different authors of the paper. The case description had a lot of grammatical errors that sometimes made it confusing or difficult to read through smoothly. The case discussion, however, was well written. I thought that this was important to point out as describing the case is almost equally as important as your discussion because other readers may be referring to the authors' case to compare their own patient. Moreover, it may deter readers from actually reading the article if it does not flow smoothly.

Reviewer #2: The authors describe a case of paracoccidioidomycosis (PCM) acquired in the state of Rio de Janeiro which was initially misdiagnosed as Chrohn’s Disease, and which lead to dissemination of PCM following immunosuppressive therapy. The authors conclude that PCM should be considered as a differential diagnosis of Chrohn’s Disease in endemic areas as it can have a similar gastrointestinal presentation. The educational aspect of this case is very interesting and it is clinically relevant. It illustrates that gastrointestinal involvement can occur not only in acute but also in chronic PCM and that a misdiagnosis as Chrohn’s Disease can potentially lead to a worse clinical trajectory. The figures are well chosen to illustrate the case. The authors review the relevant literature in this well written article. I'd like to suggest minor changes/additions. 

Title and abstract: 

The title is appropriate. The abstract is concise and focuses on the relevant information. 

Case and Discussion: 

1. Line 30: The authors mention that treatment with mesalazine and prednisone in immunosuppressive doses was prescribed. The authors should consider including the doses. 

2. Line 79: The authors cite a review of 46 cases of gastrointestinal PCM. Does the presentation of the current case, e.g. chronicity of symptoms or cutaneous lesions after starting immunosuppressive therapy, differ from the previously reported cases? The authors might consider briefly contrasting this case with a previously reported case of gastrointestinal PCM. 

3. Line 99: The authors mention that IV antifungals are usually indicated in gastrointestinal involvement as per Brazilian guidelines which was not possible in this case as the patient declined hospitalization. The authors could consider including the IV antifungals of choice, e.g. IV liposomal amphotericin B for 2-4 weeks. PMID: 28746570 

4. Line 107: The possible correlation between the patient’s residence near a highway and the increased incidence of PCM with highway construction is an interesting observation. The authors could consider mentioning the increase in PCM from 1.29 cases to 8.25 cases/1 million persons/y during a highway construction as per cited source as well as hypotheses of increased incidence due to clearing of forests and during times of increased soil humidity. PMID: 29048286. 

5. Does this potential higher risk of PCM due to residence near newly constructed highways lead to health care disparities, e.g. are people with lower income more likely to live in these areas?

PLOS authors have the option to publish the peer review history of their article (what does this mean?). If published, this will include your full peer review and any attached files.

Reviewer #1: No

Reviewer #2: Yes: Hendrik Sy

Figure Files:

Data Requirements:

Reproducibility:

References

---

## [Editor Report · Decision Letter 1]

14 Dec 2022

Dear Dr Marinho Falcão,

The authors are commended for their robust and thoughtful response to the insightful comments by the reviewers and we are therefore pleased to inform you that your manuscript 'Acute paracoccidioidomycosis worsened by immunosuppressive therapy due to a misdiagnosis of Crohn’s disease' has been provisionally accepted for publication in PLOS Neglected Tropical Diseases.

Best regards,

Joshua Nosanchuk, MD

Section Editor

.

---

## [Editor Report · Acceptance letter]

5 Jan 2023

Dear Dr Marinho Falcão,

We are delighted to inform you that your manuscript, "Acute paracoccidioidomycosis worsened by immunosuppressive therapy due to a misdiagnosis of Crohn’s disease," has been formally accepted for publication in PLOS Neglected Tropical Diseases.

Best regards,

Shaden Kamhawi

co-Editor-in-Chief

Paul Brindley

co-Editor-in-Chief
